# Alpha-1 Antitrypsin as a Regulatory Protease Inhibitor Modulating Inflammation and Shaping the Tumor Microenvironment in Cancer

**DOI:** 10.3390/cells14020088

**Published:** 2025-01-10

**Authors:** Siyu Xiang, Liu Yang, Yun He, Feng Ding, Shuangying Qiao, Zonghua Su, Zheng Chen, Aiping Lu, Fangfei Li

**Affiliations:** 1Shum Yiu Foon Shum Bik Chuen Memorial Centre for Cancer and Inflammation Research, School of Chinese Medicine, Hong Kong Baptist University, Hong Kong SAR, China; 2Institute of Precision Medicine and Innovative Drug Discovery (PMID), School of Chinese Medicine, Hong Kong Baptist University, Hong Kong SAR, China

**Keywords:** *SERPINA1*, alpha-1 antitrypsin, tumor microenvironment, inflammation

## Abstract

Alpha-1 antitrypsin (AAT) is a key serine protease inhibitor for regulating proteases such as neutrophil elastase. AAT restrains the pulmonary matrix from enzymatic degradation, and a deficiency in AAT leads to inflammatory tissue damage in the lungs, resulting in chronic obstructive pulmonary disease. Due to the crucial biological function of AAT, the emerging research interest in this protein has shifted to its role in cancer-associated inflammation and the dynamics of the tumor microenvironment. However, the lack of comprehensive reviews in this field hinders our understanding of AAT as an essential immune modulator with great potential in cancer immunotherapy. Therefore, in this review, we have elucidated the pivotal roles of AAT in inflammation and the tumor microenvironment, including the structure and molecular properties of AAT, its molecular functions in the regulation of the inflammatory response and tumor microenvironment, and its clinical implications in cancer including its diagnosis, prognosis, and therapeutic intervention. This review seeks to bridge the gap in the understanding of AAT between inflammatory diseases and cancer, and to foster deeper investigations into its translational potential in cancer immunotherapy in the future.

## 1. Introduction

The Alpha-1 Antitrypsin (AAT) protein, which is made by the *SERPINA1* gene, acts as an important inhibitor of enzymes called serine proteases and is mainly produced in the liver. AAT primarily inhibits neutrophil elastase (NE), a protease that can damage tissues during inflammatory responses. By inhibiting this enzyme, AAT helps protect tissues from excessive inflammation and tissue injury. It provides approximately 90% of the defense against the damaging effects of elastase released from neutrophils in the lower airways. Genetic mutations in the serpin superfamily, causing serpinopathies, lead to Alpha-1 Antitrypsin deficiency (AATD), which is marked by pulmonary diseases like emphysema and bronchiectasis, as well as hepatic disease [1,2].

In addition to its roles in controlling a wide range of proteolytic cascades, AAT also plays a significant role in regulating inflammation and modulating the tumor microenvironment (TME). AAT can modulate the activity of various immune cells, stromal cells, and the extracellular matrix, influencing the inflammatory response. Moreover, AAT can also reduce the production of molecules that promote inflammation, such as cytokines and chemokines, contributing to a more controlled inflammatory environment [3]. There is an urgent requirement for additional experimental and clinical investigations to fully uncover its specific role in different types of cancer and pathological conditions.

In this review, we will elucidate the pivotal roles of AAT in inflammation and the TME, including the structure and molecular properties of AAT, its molecular functions in the regulation of the inflammatory response and TME, and its clinical implications in cancer, including diagnosis, prognosis, and therapeutic intervention.

## 2. Molecular Profile of Alpha-1 Antitrypsin

### 2.1. Protein Structure of Alpha-1 Antitrypsin

The AAT protein is a serine protease inhibitor with a complex protein structure that is crucial for its function. As for its primary structure, AAT consists of 394 amino acids. The secondary structure includes approximately 60% alpha helices and around 20% beta sheets. The fully developed AAT is a glycoprotein that has three lateral chains made of carbohydrate. This 52 kDa protein consists of nine α-helices (yellow, blue, and red coils), three β-sheets (yellow, green, and red ships), and a reactive center loop (RCL), that determines its inhibitory function (Figure 1) [4,5]. In fact, this RCL domain allows for the creation of a covalent bond between the target proteinases and AAT, resulting in a stable complex that inhibits the enzyme. This transition forms the basis of the AAT inhibitory mechanism [6]. AAT typically exists as a monomer in physiological conditions. However, it can form dimers or oligomers under certain conditions, which may influence its activity and stability. A defining trait of AAT is its post-translational modification via glycosylation and oxidation. The unique structural features of AAT enable it to effectively inhibit serine proteases, particularly NE. The conformational changes it undergoes upon protease binding are crucial for its inhibitory mechanism.

### 2.2. Molecular Function of Alpha-1 Antitrypsin in Normal State

Alpha-1 Antitrypsin (AAT) is a critical protein that serves a variety of critical molecular functions, with its primary function being serine protease inhibition. AAT in its normal state achieves this by effectively inhibiting the activity of various enzymes, including protease 3, serine proteases like catalase G, and neutrophil elastase [7], acting as a protective shield against protease-driven tissue destruction. Additionally, AAT decreases neutrophil infiltration in the lungs, reduces cell death, and lowers the levels of inflammatory mediators in the plasma caused by pulmonary ischemia–reperfusion. These mediators include interleukin (IL)-1α, IL-4, IL-12p70, monocyte chemotactic protein 1, and tumor necrosis factor (TNF)-α [8]. That is to say, AAT effectively inhibits cell death and the release of inflammatory cytokines in a dose-dependent manner in vitro, and in vivo, it significantly improves lung oxygenation and mechanics while reducing lung fluid accumulation (pulmonary edema).

Furthermore, through targeted modification processes, AAT can diversify its biological functions. For instance, the introduction of sialic acid to the terminal glycans of AAT effectively protects specific residues from interactions that might compromise the stability of the molecule. Sialylated AAT exhibits a marked ability to bind to cationic chemokines, thereby inhibiting the neutralization process mediated by IL-8. This leads to a reduction in granulocyte chemotaxis through the modulation of protein–protein interactions, thus being crucial in controlling immune cell activities [9]. The glycans attached to AAT, and the resultant electrostatic charges, are instrumental in orchestrating these biological functions by facilitating the binding to the amino acid backbone of the protein. Apart from its function as a protease inhibitor, AAT is also involved in the regulation of inflammation and the modification of the TME. In addition to its protease inhibitory function, AAT also regulates inflammation and modifies the tumor microenvironment (TME).

### 2.3. Mutation of Alpha-1 Antitrypsin

The normal AAT phenotype, denoted as PI*MM, arises from the wild-type configuration of *SERPINA1*, characterized by the presence of biallelic M alleles [10,11,12]. Mutations at specific loci within the *SERPINA1* gene may lead to alterations in the protein’s conformation, resulting in either an inactive state or an aggregation-prone form, thus compromising its physiological function and contributing to disease pathogenesis [5,7,13]. Common defective variants are Z and S alleles, which often cause changes in the structure and function of AAT. The most common mutation leading to severe AATD occurs in the *SERPINA1* gene and results in the creation of the Z allele [12,14,15], which serves specific molecular functions that lead to the aggregation or buildup of misfolded proteins in the endoplasmic reticulum (ER) of innate immune cells. This results in a significant reduction in Z protein efflux from the cells, leading to ER stress, increased neutrophil apoptosis, and compromised bacterial killing. These detrimental effects culminate in excessive inflammatory responses and the manifestation of various diseases [16,17].

Individuals with the homozygous ZZ genotype experience a significant decrease in circulating plasma AAT levels, dropping to fewer than 10% of the normal protein concentrations. This deficiency leads to a twofold increase in the time required to inhibit NE compared to individuals with the M variant of AAT (M-AAT), indicating significantly compromised protease inhibition. Protein aggregates of Z-AAT have been linked to the development of cirrhosis and chronic hepatitis. The buildup of Z-AAT in liver cells causes problems with protein secretion, leading homozygous Z mutation carriers to have only 10–15% of the usual levels of AAT. AAT deficiency triggers a substantial influx of neutrophils into the airways, the heightened release of serine proteases, and uncontrolled NE activity, ultimately culminating in lung parenchymal damage, emphysema, and chronic obstructive pulmonary disease (COPD) [18].

On the other hand, alleles like the S variant have a relationship with a milder form of AATD: the S mutation is known to form aggregates, although at a rate that is lesser than the Z mutation. This difference leads to less retention of substances in liver cells and prevents liver disease. Despite some of the misfolded S protein being eliminated by specific cellular processes, a fraction of the S protein is correctly folded and released into the bloodstream. As a result, the serum AAT levels fall within an intermediate range between the PI*MM and PI*ZZ phenotypes [12,19,20]. The presence of these alleles with pathogenic mutations from other alleles has been demonstrated to increase the risk of AATD. Due to these mutations, individuals with AATD might have lower levels of AAT in their blood and tissues, especially in their lungs.

## 3. Regulation of Inflammatory Responses by Alpha-1 Antitrypsin

AAT is a key factor in the inflammatory response, regulating inflammation mainly through interacting with proteases, affecting cytokines, regulating cell apoptosis, and interacting with the complement system(Table 1).

### 3.1. Inhibition of Proteases

AAT can interact with proteases, especially neutrophil elastase (NE), to regulate inflammation. Research conducted by Cai M et al. [21] showed that AAT inhibited the formation of neutrophil extracellular traps by acting on NE, which helped protect mice from inflammation and coagulation caused by sepsis. The anti-inflammatory and immune-modulating effects of AAT can occur without involving NE [34]. The research conducted by Geraghty P et al. [3] showed that AAT activates Protein Phosphatase 2A to stop the inflammatory and protein-degrading responses caused by TNF-α stimulation in the lungs. Cigarette smoke causes oxidative stress and cellular damage, while AAT elicits anti-inflammatory effects by binding to and decreasing cigarette smoke-induced endothelial sheddases, thrombin, and plasmin in the airways [22,23,24]. It has been suggested that AAT may be used to lower lung inflammation by fine-tuning the CX3CL1-CX3CR1 axis, which is specifically implicated in endothelial–monocyte crosstalk and leukocyte migration to the alveolar region. Furthermore, AAT also reduces inflammation and exerts chondroprotection in arthritis. AAT decreased the expression of the genes mmp13 and adamts5 and increased the transcription of col2a1, acan, and sox9 in ex vivo studies of arthritic joints [35]. These studies demonstrate that AAT aids in anti-inflammatory activities by the inhibition of proteases.

### 3.2. Effect on Cytokine Production

AAT might have biological effects apart from its role as a protease inhibitor. It has been demonstrated that AAT affects the synthesis of a number of pro-inflammatory cytokines. By inhibiting the activity of these cytokines, AAT can reduce inflammation. AAT emerges as a pivotal regulator in the modulation of cytokine synthesis and release, potentially suppressing inflammatory responses. Accumulating evidence suggests AAT stands out as the most abundant endogenous inhibitory component for inflammation mediators within human circulatory systems.

Janciauskiene S et al. [25] demonstrated that AAT has anti-inflammatory properties that are independent of the inhibition of serine proteases in vitro. AAT can inhibit lipopolysaccharide (LPS)-mediated human monocyte activation, leading to the decreased production of pro-inflammatory cytokines and chemokines. Moreover, ATT inhibits LPS-induced NFκB activation and IL-8 production in macrophages by binding to the glucocorticoid receptor, providing anti-inflammatory and anti-mycobacterial effects [36]. Herr C et al. [26] showed that the distinct patterns of AAT and cytokines (IL-1Ra, IL-8, and IL-10) underscore the complexity of the immune response in COVID-19. AAT plays a protective role by inhibiting excessive inflammation and tissue damage. Its presence can help regulate the immune response, potentially reducing the severity of COVID-19 and improving survival rates.

Through both ex vivo and in vitro observations, Pott GB et al. [27] came to the conclusion that endogenous AAT in blood can help reduce the manufacture of pro-inflammatory cytokines. Exogenous AAT strongly suppressed IL-1β, IL-6, IL-8, and TNF-α in whole blood that was diluted and challenged with *Staphylococcus epidermidis*, but it did not suppress the generation of the anti-inflammatory cytokines IL-10 and IL-1R antagonist (IL-1Ra). Mice treated with AAT had reduced levels of inflammatory cytokines in their blood right after the transplant [28]. Despite both being potent anti-inflammatory drugs, AAT and corticosteroids work in different ways. AAT mainly controls cytokine synthesis and inhibits proteolytic enzymes to modify immune responses. On the other hand, corticosteroids primarily work through genomic processes, affecting gene expression and encouraging the production of anti-inflammatory proteins [33].

By combining the complete AAT gene with the constant portion of IgG1 to produce the soluble recombinant AAT-Fc protein, Lee S et al. [37] evaluated the anti-inflammatory qualities of AAT. They demonstrated that AAT-Fc prevented the production of TNF-α triggered by various cytokines in mouse macrophage Raw 264.7 cells and also reduced the rise in IL-6 levels caused by TNF-α in human peripheral blood mononuclear cells. This suggests that AAT-Fc demonstrates a robust capacity to reduce the production of pro-inflammatory cytokines. This new recombinant protein highlights the potential of using modified AAT variations as powerful modulators in the coordination of immune responses, offering a possible treatment option for pathological disorders caused by inflammation.

### 3.3. Regulation of Apoptosis

Apoptosis, a crucial form of programmed cell death, is vital for maintaining tissue homeostasis and eliminating damaged or superfluous cells. On the one hand, AAT can regulate antiapoptotic activated immune cells, reducing the overall production of inflammatory cytokines and mediators. In those who are lacking AAT, augmentation therapy with AAT can reverse increased neutrophil apoptosis and decrease TNF-α signaling [16]. Aldonyte R et al. [29] showed that exogenous AAT could be collected by pulmonary arterial endothelial cells and protected cells from cigarette smoke-induced apoptosis. The results reported by Petrache I et al. broadened the lung protective roles of AAT beyond its effects on NE, to include apoptosis blockade and implicated an antiapoptotic effect of AAT in alveolar cells [38]. Additionally, AAT dramatically improved lung oxygenation and lung mechanics, decreased pulmonary edema in vitro, and dose-dependently decreased cell death and inflammatory cytokine release in vitro, thereby inhibiting ischemia–reperfusion-induced lung injury [8].

On the other hand, ATT can directly affect the apoptosis of cancer cells. The interaction between AAT and tumor progression underscores the complexity of the tumor microenvironment, where inflammatory responses can either favor or inhibit cancer development [39]. Cancer cells may produce or uptake exogenous AAT to exploit its anti-inflammatory and cytoprotective properties, which can enhance their survival and proliferation. AAT’s capacity to regulate immune responses to inhibit apoptosis suggests that it could provide cancer cells a selective advantage, allowing them to thrive in hostile environments. According to the results from Schwarz N et al. [40], AAT contributes significantly to carcinogenesis and shields lung cancer cells from staurosporine-induced apoptosis (programmed cell death). Additionally, the study suggested that bacterial lipopolysaccharide (LPS) plays a role in this protective effect, possibly by influencing the activity or expression of AAT via the inhibition of AKT/MAPK pathways and the activation of caspase 3 and autophagy. The results presented by Petrache I et al. illustrated that blocking active caspase-3 directly with AAT could be a new way to prevent cell death, which may be important for diseases where too many cells are dying, and there is a lot of oxidative stress and inflammation [30]. Conversely, AAT therapy by Al-Omari et al. [41] showed promising potential in mouse models of colitis-associated colon cancer by exerting anti-inflammatory effects, protecting tissue integrity, modulating immune responses, reducing tumorigenesis, and enhancing the apoptosis of cancer cells. Therefore, gaining deeper insights into how AAT interacts with inflammation and cancer cell resistance to programmed cell death is very important for clinical research and treatment.

### 3.4. Interactions with the Complement System

An integral part of the innate immune response is the complement system, sometimes referred to as the complement cascade, consisting of a series of small proteins found in circulatory systems and primarily produced by hepatic cells. When activated, this mechanism improves the efficacy of antibodies and phagocytes in removing germs and damaged cellular elements, promoting an inflammatory milieu, and launching an attack on pathogen cellular membranes. Enzyme inhibitors, such as AAT, limit the activation of the complement system, the primary humoral mediator of inflammation.

O’Brien, M. E. et al. highlighted the immune-modulatory impact of AAT on the complement system. Complement C3 was found to directly bind to AAT both in vitro and in vivo. A breakdown product of C3, C3d, was elevated in AATD compared to healthy controls, and there was a significant connection between the plasma levels of C3d and radiographic lung emphysema (R^2^ = 0.37, *p* = 0.001). They also found that AAT augmentation therapy dramatically reduced C3d plasma levels in vivo [31]. Moreover, research conducted by Gou, W. et al. showed that following intrahepatic islet transplantation, AAT increased islet graft survival and inhibited macrophage activation. They suggested this potential AAT therapy could reduce M1 macrophage polarization and cytokine-induced damage. These cells cause damage to other cells by secreting inflammatory cytokines and free radicals, such as IL-1, IL-6, TNF-α, IFN-γ, coagulation factors, complement, reactive oxygen species, and nitrogen [28]. This emphasizes the importance of future investigation into the connections between AAT and the complement system to uncover further aspects of the anti-inflammatory properties of AAT.

## 4. Influence of Alpha-1 Antitrypsin on Tumor Microenvironment

TME plays an essential part in the neoplastic process by promoting cell migration, survival, and proliferation. It is primarily influenced by the inflammatory response. Several studies in recent years have found connections between AAT levels and the likely outcome or progression of cancer; however, the findings are conflicting. It has become more obvious that cancer patients with high plasma levels of AAT have a poor prognosis. AAT also can modulate inflammatory responses within the TME. First, AAT can help regulate the activity of neutrophils and other immune cells by inhibiting proteases, reducing tissue damage and chronic inflammation, which can promote tumor progression in many types of cancer. Second, AAT can affect macrophages and T cells. It may promote a stronger anti-tumor immune response by polarizing macrophages toward an M2 phenotype, which is frequently linked to inflammation reduction and tissue restoration. Third, AAT can affect tumor cells by regulating the extracellular matrix (ECM) within the TME. In addition, AAT’s effect on inflammatory mediators can influence the angiogenic process, which is crucial for tumor growth and metastasis (Table 2).

### 4.1. AAT Can Help Regulate the Activity of Neutrophils in TME

Neutrophils, being part of the tumor microenvironment (TME), are believed to significantly influence tumor development and the ability of the tumor to invade surrounding tissues. Zelvyte, I. et al. showed that factors produced from neutrophils increase cell invasiveness while decreasing lung cancer cell proliferation and IL-8 release. An exogenous serine proteinase inhibitor, AAT, and its C-terminal fragment were demonstrated to alter these effects, demonstrating the complexity of the relationships between the biological activity of tumor cells and the surrounding microenvironment [32]. According to Xu et al., in the inflammatory microenvironment, NE and AAT are crucial for controlling the growth of lung tumors. Curcumin increases AAT expression both in vitro and in vivo, which inhibits NE-induced tumor growth [42].

AAT controls the movement of neutrophils by attaching to IL-8, preventing IL-8 from connecting with its receptor, CXCR1, on the neutrophil surface. It was found that neutrophils move in the opposite direction of an AAT gradient when there is a rising IL-8 gradient. The process is significantly influenced by the glycosylation status of AAT, which underscores its immunomodulatory efficacy [43]. David A. Bergin et al. illustrated that serum AAT manages the internal signaling of TNF-α and the release of granules from neutrophils by influencing the ligand–receptor interaction. Its ability to modulate inflammatory responses highlights its importance in maintaining immune homeostasis in COPD [60].

### 4.2. AAT May Promote a Stronger Anti-Tumor Immune Response by Polarizing Macrophages Toward an M2 Phenotype

Macrophages play an integral role within the tumor immune microenvironment, engaging in complex interactions with an array of immune cells, such as dendritic cells and natural killer cells. Our earlier studies on classifying osteosarcoma (OS) based on inflammatory genes showed that inflammation in the tumor microenvironment (TME), driven by myeloid-derived immune cells like macrophages, is linked to a better prognosis [44]. Macrophages within the tumor milieu exhibit a spectrum of phenotypes, extending from pro-inflammatory (M1) to anti-inflammatory (M2). The first group is associated with killing pathogens inside cells and resisting tumors, while the second group is involved in regulating the immune system, promoting the growth of new blood vessels (which helps tumors grow), and suppressing the adaptive immune response [12,33].

Gou, W. et al. discovered that AAT prevented the polarization of M1 macrophages induced by IFN-γ by blocking the phosphorylation of STAT1. Notably, when liver macrophages were removed, it was shown that M1 macrophages in the liver played a role in the failure of transplanted islets to function properly. AAT stops macrophage activation triggered by cytokines or dying islets, thereby promoting the survival of islet cells [28]. According to Fu, C. et al., changes in DNA methylation, gene mutations, and modifications made to *SERPINA1* after it is produced were all found to be strongly and positively associated with its expression levels in breast cancer, making it a potential diagnostic marker. They additionally identified that the *SERPINA1* gene relates positively with macrophages and can trigger M2 macrophage polarization. *SERPINA1* was found to be related to macrophages in glioma immunological microenvironments [61].

Through a LASSO logistic regression analysis and multi-factor COX regression analysis, Wu L et al. [45] discovered that the *SERPINA1* gene, encoding AAT protein, emerges as one of the central upregulated genes associated with lymphatic metastasis and one of the independent prognostic factors in thyroid carcinoma (THCA). However, an immuno-infiltration analysis showed that the expression of the *SERPINA1* gene was found to be significantly favorably related with M1 macrophages and NK cells, but negatively associated with T cell CD4+ and myeloid dendritic cells. This implies that *SERPINA1* is critical for immune cell infiltration in THCA.

Abnormal glycosylation alterations are known to influence malignant tumor cell proliferation and migration while simultaneously increasing tumor-induced immune regulatory responses [46]. Microglia, or central nervous system tumor-associated macrophages, are a critical component of the glioma TME, capable of producing a variety of growth factors and cytokines to modify the glioma immune landscape [62]. Several studies have highlighted that glycosylation-related differentially expressed genes are significantly associated with glioma progression. Comprehensive bioinformatics investigations have identified the *SERPINA1* glycosylation gene as one of seven glycosylation genes associated with glioma prognosis, indicating its involvement in cytokine signaling, inflammatory responses, immunological control, glycan synthesis, and metabolic pathways. As glioma advances, the *SERPINA1* glycosylation gene contributes to the heightened infiltration of diverse immune cells, including macrophages [63].

### 4.3. Influence of AAT on T Cells

AAT has the ability to modulate the immune system in various immune disorders that depend on T cells. By suppressing natural killer cell activity, T cell-mediated cytotoxicity, and antibody-dependent cell-mediated cytotoxicity, AAT may modify host immunodefense systems to support tumor cells, thereby reducing lymphocyte blastogenic or cytotoxic responses [47,48,49]. CD8+ T cells play a crucial role in tumor elimination by directly killing tumor cells and increasing local IFNγ levels, which promote tumor-suppressing Th1 responses [50,51,52]. T cell responses are facilitated by the highly selective immunomodulator AAT. Guttman, O. et al. identified that AAT inhibits the growth of tumors in a way that is dependent on CD8+ T cells. Upon analyzing the tumor cellular composition, they also showed M1-like TAMs and functional tumor-infiltrating CD8+ T cells in mice treated with AAT [49].

Bristow, C. L. et al. showed that AAT augmentation dramatically raised the CD4/CD8 ratio in both HIV-1-positive and -uninfected people. AAT regulated the transformation of CD4+CD8+ T cells into immunologically competent CD4+ T cells [53]. By default, CD4+CD8+ double positive (DP) T cells mature to become CD8+ single positive (SP) T cells. Signaling induced by the binding of AAT to human leukocyte elastase on the cell surface of DP T cells induces NFκB phosphorylation and stimulates the maturation of DP T cells to become CD4+ SP T cells. AAT causes CD8 downregulation during thymocyte differentiation [64].

### 4.4. AAT Effect on Inflammatory Mediators Can Influence the Angiogenic Process

Endothelial cells line the inside of blood vessels and play a crucial role in angiogenesis, the formation of new blood vessels. In the early stages of cancer, tumor cells depend on passive diffusion for nutrient acquisition and gas exchange. However, as the tumor grows and seeks to spread, it induces angiogenesis to develop a new vascular supply. The AAT protein influences angiogenesis, which affects protease activity and inflammatory responses, key factors in the angiogenic process. *SERPINA1* specifically modulates serine protease activity, reducing pro-angiogenic and pro-inflammatory responses when PAR-2 is activated in endothelial cells [65]. In gastrointestinal diseases, increased *SERPINA1* expression is linked to better endothelial cell survival, vascular abnormalities, new vessel growth, and higher vascular permeability [54]. *SERPINA1* plays a key role in the communication between endothelial and tumor cells within the TME, impacting various aspects of cancer development and progression [55]. Thus, *SERPINA1* is integral to the crosstalk between endothelial cells and neoplasms, influencing myriad facets of cancer pathogenesis and evolution.

### 4.5. AAT Can Affect Tumor Cells by Regulating the Extracellular Matrix (ECM) Within the TME

The ECM is a complex network comprising fibrous proteins (collagens and elastin) and glycoproteins (fibronectin, proteoglycans, and laminins) that serves as a critical scaffold for cellular constituents and significantly influences the TME [66,67]. This matrix is pivotal in directing tumor cell behavior, including proliferation, invasion, and metastasis, while protecting tumor cells from apoptosis induced by chemotherapeutic drugs, leading to drug resistance and tumor resurgence post-treatment [68]. There are various biochemical and structural types of ECM. Specific signals are conveyed to cells by each form and its three-dimensional structure, affecting vital processes such as immune cell migration into inflammatory tissues and immune cell differentiation in the early phases of inflammation [69,70].

AAT can regulate collagen synthesis and deposition by preventing collagen-degrading proteases, modifying fibroblast activity, and decreasing inflammation. These impacts on collagen can have a major impact on tumor cell behavior by affecting the physical and biochemical properties of the extracellular matrix, influencing cell adhesion, migration, invasion, and overall tumor growth [56,57,58].

AAT can inhibit various proteases like matrix metalloproteinases (MMPs), which are involved in ECM degradation. By inhibiting these proteases, AAT can reduce ECM remodeling, which is often necessary for tumor invasion and metastasis. Churg, A. et al. [24] suggested that AAT blocks cigarette smoke and thrombin-dependent activation of TNF-α and Matrix Metallopeptidase (MMP)-12 in alveolar macrophages. Smoke-conditioned medium induces alveolar macrophages to secrete higher levels of macrophage MMP-12 and pro-inflammatory cytokine TNF-α, but this effect is inhibited in a dose-dependent manner by the addition of AAT. Additionally, macrophages from animals exposed to smoke in vivo did not show increased secretion of MMP-12 and TNF-α when the animals were pretreated with AAT.

Tuder, R.M. and I. Petrache [59] showed that AAT encoded by *SERPINA1* may prevent the activation of alveolar macrophages by neutralizing certain enzymes in the coagulation process, especially thrombin and plasmin. This stops the activation of protease-activated receptors (PARs) caused by either cigarette smoke or thrombin. AAT could help regulate this process while also protecting the lungs in other ways [35]. AAT can affect tumor cells by regulating the ECM within the TME through its protease inhibitory activity, the modulation of ECM components, and anti-inflammatory effects. The potential for these ECM changes to impact tumor cell invasion, metastasis, and immune cell accessibility to tumor cells highlights the complex function of AAT in cancer pathophysiology.

## 5. Clinical Implications of AAT for Cancer

### 5.1. Biomarker for Cancer Diagnosis and Prognosis

Considering the implications of AAT on inflammation and immune responses, the levels of AAT and the *SERPINA1* gene in blood or tissue samples have garnered significant interest as potential biomarkers for the diagnosis or prognosis of certain oncological conditions. Extensive research underscores the utility of AAT and the *SERPINA1* gene as prognostic or diagnostic indicators in a variety of cancer types, as demonstrated by a comprehensive body of research (Table 3). The promising role of AAT in cancer diagnosis and prognosis heralds innovative avenues for personalized therapeutic strategies.

### 5.2. Potential Target for Therapeutic Intervention

Alpha-1 Antitrypsin (AAT) and the *SERPINA1* gene are emerging as promising therapeutic targets in a range of malignancies.

Research indicates that *SERPINA1* serves as a potential target for miR-486-3p, providing a novel approach for the targeted therapy of invasive thyroid carcinoma [85]. Moreover, an investigation by Luo, L. et al. [86] delineated *SERPINA1* as a pivotal gene in breast cancer pathogenesis through a bioinformatics analysis, subsequently identifying prospective drugs that may broaden treatment paradigms and facilitate the exploration of new avenues for breast cancer therapy. Moreover, Li, W. et al. [87] underscored the critical importance of *SERPINA1* as a therapeutic target in bladder cancer management, further emphasizing its significance across different malignancies. Through a comprehensive bioinformatics examination, *SERPINA1* has also been identified as a potential therapeutic target for osteosarcoma, underscoring the versatility and breadth of its applicability in cancer treatment [84]. Furthermore, Zhang, L. et al. [88] highlighted the significant role of *SERPINA1* in the liver metastasis of colon cancer, probing novel target areas for the amelioration of colon cancer outcomes. Beyond its association with specific cancer types, AAT has been postulated as a novel intervention point for the treatment of tumor metastasis in lung cancer [89]. These findings suggest a promising horizon for the utilization of AAT and *SERPINA1* as therapeutic targets.

## 6. Conclusions and Future Perspective

AAT is pivotal in the modulation of inflammatory expression and the maintenance of homeostasis within the human body. Research has elucidated the genetic codification, allelic variants, structural composition, and molecular operations of AAT, consequently improving the comprehension of its integral part in the inflammatory cascade.

AAT primarily influences the inflammatory response through mechanisms such as protease interactions, cytokine interplay, apoptosis regulation, and engagement with the complement system. Concurrently, *SERPINA1* is recognized for its regulatory influence over various constituents within the TME, encompassing immune and cancer cells, as well as the extracellular matrix. Hence, both AAT and the *SERPINA1* gene could be promising markers for cancer diagnostics and prognosis, as well as potential focal points for oncological therapeutics.

## Figures and Tables

**Figure 1 cells-14-00088-f001:**
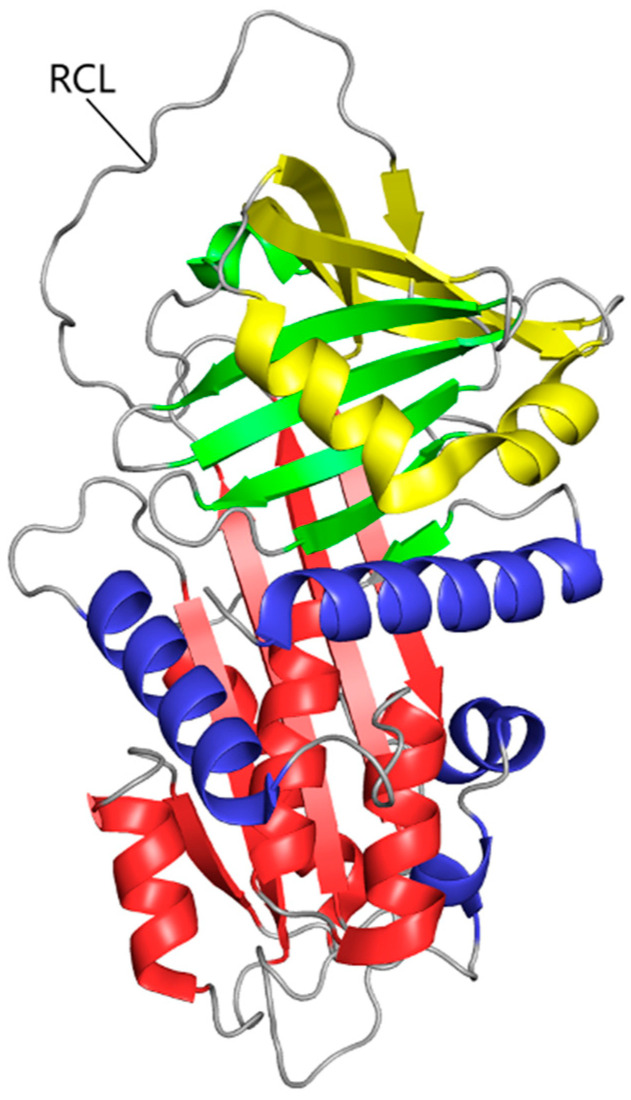
The tertiary structure of AAT. The polypeptide chain is composed of nine α-helices (yellow, blue, and red coils), three β-sheets (yellow, green, and red ships), and a reactive center loop.

**Table 1 cells-14-00088-t001:** Regulation of inflammatory responses by Alpha-1 Antitrypsin through interacting with proteases, affecting cytokines, regulating cell apoptosis, and interacting with the complement system.

Modes of Action	Mechanism	
Inhibition of proteases	*Neutrophil elastase (NE)*	AAT inhibited the formation of neutrophil extracellular traps by acting on NE, which helped protect mice from inflammation and coagulation caused by sepsis [21].
	*Protein Phosphatase 2A*	AAT activates Protein Phosphatase 2A to stop the inflammatory and protein-degrading responses caused by TNF-α stimulation in the lungs [3].
	*Endothelial sheddases, thrombin and plasmin*	The Helicobacter pylori IgG level was higher in PDAC Helicobacter pylori may promote the development of PDAC by causing chronic mucosal inflammation as well as changes in cell proliferation and differentiation [22,23,24,25].
Effect on cytokine production	ATT inhibits LPS-induced NFκB activation and IL-8 production in macrophages providing anti-inflammatory and anti-mycobacterial effects [26]. AAT and cytokines (IL-1Ra, IL-8, IL-10) underscore the complexity of the immune response in COVID-19 [27]. Exogenous AAT strongly suppressed IL-1β, IL-6, IL-8, and TNF-α in whole blood [28]. AAT-Fc prevented the production of TNFα triggered by various cytokines in mouse macrophage Raw 264.7 cells and also reduced IL-6 levels caused by TNF-α in human peripheral blood mononuclear cells [29].	
Regulation of apoptosis	Immune cells anti-apoptotic	Reducing the overall production of inflammatory cytokines and mediators Augmentation therapy with AAT can reverse increased neutrophil apoptosis and decrease TNF-α signaling, in those who are lacking [16].
	Apoptosis of cancer cells	AAT contributes significantly to carcinogenesis and shields lung cancer cells from staurosporine-induced apoptosis [30]. Conversely, AAT therapy by Al-Omari et al. showed promising potential in mouse models of colitis-associated colon cancer by exerting anti-inflammatory effects, protecting tissue integrity, modulating immune responses, reducing tumorigenesis, and enhancing apoptosis of cancer cells [31].
Interactions with the complement system	Complement C3 was found to directly bind to AAT both in vitro and in vivo. A breakdown product of C3, C3d, was elevated in AATD compared to healthy controls [32]. AAT increased islet graft survival and inhibited macrophage activation. The potential therapy to reduce M1 macrophage polarization and cytokine-induced damage through AAT treatment These cells cause damage to other cells by secreting inflammatory cytokines and free radicals, such as IL-1, IL-6, TNF-α, IFN-γ, coagulation factors, complement, reactive oxygen species, and nitrogen [33].	

**Table 2 cells-14-00088-t002:** Influence of Alpha-1 Antitrypsin on tumor microenvironment.

The Components	Mechanism (*SERPINA1*, Encoded AAT)
Neutrophils	**AAT can help regulate the activity of neutrophils in TME** Curcumin increases AAT expression both in vitro and in vivo, which inhibits NE-induced tumor growth [42,43]. AAT controls the movement of neutrophils by attaching to IL-8, preventing IL-8 from connecting with its receptor, CXCR1, on the neutrophil surface [44].
Macrophages	**AAT may promote a more anti-tumor immune response by polarizing macrophages toward an M2 phenotype** AAT prevented the polarization of M1 macrophages induced by IFN-γ by blocking the phosphorylation of STAT1 [33]. *SERPINA1* gene relates positively with macrophages and can trigger M2 macrophage polarization [45]. *SERPINA1* gene was found to be significantly favorably related with M1 macrophages and NK cells in thyroid carcinoma [46]. As glioma advances, the *SERPINA1* glycosylation gene contributes to heightened infiltration of diverse immune cells, including macrophages [47].
T cells	**AAT can modulate the immune system in various immune disorders that depend on T-cells.** AAT may modify host- immunodefense systems to support tumor cells, thereby reducing lymphocyte blastogenic or cytotoxic responses by suppressing natural killer-cell activity, T cell mediated cytotoxicity, and antibody-dependent cell-mediated cytotoxicity [48,49,50]. T cell responses are facilitated by the highly selective immunomodulator AAT [51,52,53]. M1-like TAMs and functional tumor- infiltrating CD8+ T-cells in mice treated with AAT [50].
Endothelial cell	**AAT influences angiogenesis, which affects protease activity and inflammatory responses, key factors in the angiogenic process** *SERPINA1* specifically modulates serine protease activity, reducing pro-angiogenic and pro-inflammatory responses when PAR-2 is activated in endothelial cells [54]. Increased *SERPINA1* expression is linked to better endothelial cell survival, vascular abnormalities, new vessel growth, and higher vascular permeability [55].
Extracellular matrix	**AAT can affect the remodeling of ECM and cooperate with its components to drive the migration and growth of cancer cells and affect further progression.** **Collagen:** AAT can regulate collagen synthesis and deposition by preventing collagen-degrading proteases and modifying fibroblast activity [56,57,58]. **MMPs:** AAT can inhibit various proteases like matrix metalloproteinases (MMPs), which are involved in ECM degradation [24]. **Pro-inflammatory cytokines:** AAT can decrease the levels of pro-inflammatory cytokines TNF-α within the TME [59]. **Thrombin and plasmin:** AAT may prevent the activation of alveolar macrophages by neutralizing certain enzymes in the coagulation process, especially thrombin and plasmin [35,59].

**Table 3 cells-14-00088-t003:** Tumor types in which AAT or *SERPINA1* show potential as biomarkers.

Type	Reference
Gastric cancer	[71,72]
Lung cancer	[73,74]
Pancreatic cancer	[75]
Colorectal cancer	[76,77,78]
Papillary thyroid carcinoma	[79]
Human glioblastoma	[80]
Kidney cancer	[72]
Prostate cancer	[81]
Oral cancer	[82]
Bladder cancer	[83]
Osteosarcoma	[84]

## Data Availability

No new data were created or analyzed in this study.

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
