# Peer review of "Alpha-1 Antitrypsin as a Regulatory Protease Inhibitor Modulating Inflammation and Shaping the Tumor Microenvironment in Cancer"

_cells, 2025, doi:10.3390/cells14020088_

Round 1

Reviewer 1 Report

Comments and Suggestions for Authors

The authors describe alpha-1-antitrypsin (AAT) as a key serine protease inhibitor involved in regulating proteases such as neutrophil elastase. AAT safeguards the pulmonary matrix from enzymatic degradation, and its deficiency can lead to inflammatory damage in lung tissue, resulting in conditions like chronic obstructive pulmonary disease (COPD). However, the authors emphasize a growing interest in AAT's role in cancer-associated inflammation and its impact on tumor microenvironment dynamics.

The review focuses on the following aspects:

  1. Structure and molecular properties of AAT: Highlighting how these relate to its function.
  2. Molecular functions of AAT: In regulating inflammatory responses and interacting with the tumor microenvironment.
  3. Clinical implications in cancer: Exploring its potential in diagnosis, prognosis, and therapeutic interventions.

This study underscores AAT's translational potential in oncology and aims to bridge the gap in understanding between its roles in inflammatory diseases and cancer.

However, there are things that would improve the work:

Figures:
I believe additional illustrative figures would enhance the text's clarity, ideally including at least one per section. For instance, in Section 4, it would be helpful to add an image that outlines all the components involved in modifying the tumor microenvironment.

Regarding Figure 2, it appears overly simplistic. Additionally, the disproportionately large size of the NPS overshadows the other elements in the diagram, making it harder to grasp the overall concept.

Text Errors:

  • In line 210, the text is formatted in a different size compared to the rest, which disrupts the document's uniformity.
  • In line 390, there is a citation error: (Error! Reference source not found.). This issue should be corrected to ensure consistent referencing and avoid reader confusion.

These adjustments will improve both the visual clarity and the overall coherence of the manuscript.

Author Response

Comment 1: "I believe additional illustrative figures would enhance the text's clarity, ideally including at least one per section. For instance, in Section 4, it would be helpful to add an image that outlines all the components involved in modifying the tumor microenvironment."

Response 1: Thanks for pointing this out. we agree with this comment, and updated Figures and tables.

Comment 2: "Regarding Figure 2, it appears overly simplistic. Additionally, the disproportionately large size of the NPS overshadows the other elements in the diagram, making it harder to grasp the overall concept."

Response 2: Thanks for pointing this out. we agree with this comment, and updated Figure2.

Comment 3: "In line 210, the text is formatted in a different size compared to the rest, which disrupts the document's uniformity. In line 390, there is a citation error: (Error! Reference source not found.). This issue should be corrected to ensure consistent referencing and avoid reader confusion."

Response 3: Thanks for pointing this out. We have made changes.

Reviewer 2 Report

Comments and Suggestions for Authors

The review article entitled: Alpha-1 Antitrypsin as a regulatory protease inhibitor modulating inflammation and shaping the tumor microenvironment in cancer, by Siyu XIANG et al., is focusing on the pivotal roles of alpha -1 antitrypsin (AAT) in inflammation and tumor microenvironment including its structure and molecular properties. It depicts  the involvement and impact on  AAT in inflammation and cancer.

The question raised here is what is the benefit gained by another review article on this subject. Unfortunately, it contains multiple statements without in-depth explanations. It feels as the authors incorporated key sentences of the  relevant articles from the abstracts and titles. It is redundant to read repeatedly such sentences without a comprehensive description in the field.

- For example, in line 218,  the work by Schwartz N et al., is described as follows:

 "Schwarz N et al., AAT protects lung cancer cells from staurosporine-induced apoptosis and plays a significant role in the tumorigenesis[37]. This is the only sentence describing the work of Schwartz. In-fact, the manuscript by Natalie Schwarts describes the molecular machinery of AAT. It acts via inhibition of AKT/MAPK pathways, and activation of caspase 3 and autophagy.

- In addition, section 4 on tumor microenvironment, the sentence that AAT influences TME constituents is continually mentioned. Rather one should illustrate which of the components are affected and describe relevant examples appearing in  the literature! It is not acceptable to write (line 263): …. AAT can affect tumor  cells by regulating the extracellular matrix (ECM) within the TME. This is by no means sufficient.

-While the title calls for the involvement of AAT in cancer, there is no mention of a specific effect of AAT on cancer cells or receptors which reside on cancer cells.

- The review completely ignores the impact of AAT, as a protease inhibitor,  on  protease -activated receptors ( PARs)  as well as on coagulation proteins. They should cite the nice review by Tuder RM and Petrache I on: Molecular Multitasking in the Airspace_1-Antitrypsin Takes on Thrombin and Plasmin, describing comprehensively the impact of AAT on lung and hepatic cancers.

-The part on angiogenesis is shallow and not sufficient! It is not clear why they cite a paper on PAR2 and TNF-alpha,  impacting on Tie2. There is  no relation to AAT.

- It is required to present a Table summarizing the role of AAT in inflammation.

-It is required to show a Table on the role of AAT on cancer cells.

Author Response

Comment 1: "For example, in line 218,  the work by Schwartz N et al., is described as follows: Schwarz N et al., AAT protects lung cancer cells from staurosporine-induced apoptosis and plays a significant role in the tumorigenesis[37]. This is the only sentence describing the work of Schwartz. In-fact, the manuscript by Natalie Schwarts describes the molecular machinery of AAT. It acts via inhibition of AKT/MAPK pathways, and activation of caspase 3 and autophagy."

Response 1: Thanks for pointing this out. we agree with this comment. We have made changes.

AAT contributes significantly to carcinogenesis and shields lung cancer cells from staurosporine-induced apoptosis(programmed cell death). Additionally, the study suggested that bacterial lipopolysaccharide (LPS) plays a role in this protective effect, possibly by influencing the activity or expression of AAT via inhibition of AKT/MAPK pathways, and activation of caspase 3 and autophagy. 

Comment 2: In addition, section 4 on tumor microenvironment, the sentence that AAT influences TME constituents is continually mentioned. Rather one should illustrate which of the components are affected and describe relevant examples appearing in  the literature!

Response 2: Thanks for pointing this out. We elaborated on each of these in section 4: for example, 4.1. AAT can help regulate the activity of neutrophils in TME; 4.2. AAT may promote a more anti-tumor immune response by polarizing macrophages toward an M2 phenotype; 4.3. T cells; 4.4. Endothelial cells; 4.5. Extracellular matrix.

Comment 3: While the title calls for the involvement of AAT in cancer, there is no mention of a specific effect of AAT on cancer cells or receptors which reside on cancer cells.

Response 3: Thanks for pointing this out. In response to this, we mainly discussed the influence of AA T on tumor microenvironment and how AAT affects tumor progression by affecting various elements in TME (section 4).

Comment 4: The review completely ignores the impact of AAT, as a protease inhibitor,  on  protease -activated receptors ( PARs)  as well as on coagulation proteins. They should cite the nice review by Tuder RM and Petrache I on: Molecular Multitasking in the Airspace_1-Antitrypsin Takes on Thrombin and Plasmin, describing comprehensively the impact of AAT on lung and hepatic cancers.

Response 4: Thanks for pointing this out. we have read this journal paper by Tuder, R.M. and I. Petrache and learned a lot and cited this paper for PARs about AAT. 

Comment 5: The part on angiogenesis is shallow and not sufficient! It is not clear why they cite a paper on PAR2 and TNF-alpha,  impacting on Tie2. There is  no relation to AAT.

Response 5: Thanks for pointing this out. “SERPINA1 specifically modulates serine protease activity, reducing pro-angiogenic and pro-inflammatory responses when PAR-2 is activated in endothelial cells.” AAT is encoded by SERPINA1 gene.

Comment 5: It is required to present a Table summarizing the role of AAT in inflammation and cancer.

Response 5: Thanks for pointing this out. We have made changes.(Table1-2)

Round 2

Reviewer 2 Report

Comments and Suggestions for Authors

Still in the revised manuscript, the authors have not addressed adequately issues raised in point 2. 

What components of the ECM are affected ? It is not sufficient to  say that AAT affects ECM-degrading proteases. It should be explained more in -depth in the paragraph. It is good to see in the Table citation of some references, however it does not help the reader to understand the involvement of ECM, for example. As per the response to point 2, it applies also to endothelial cells.

Author Response

comment 1:

round 1: "In addition, section 4 on tumor microenvironment, the sentence that AAT influences TME constituents is continually mentioned. Rather one should illustrate which of the components are affected and describe relevant examples appearing in  the literature! It is not acceptable to write (line 263): …. AAT can affect tumor  cells by regulating the extracellular matrix (ECM) within the TME. This is by no means sufficient."

round 2: "Still in the revised manuscript, the authors have not addressed adequately issues raised in point 2. 

What components of the ECM are affected ? It is not sufficient to  say that AAT affects ECM-degrading proteases. It should be explained more in -depth in the paragraph. It is good to see in the Table citation of some references, however it does not help the reader to understand the involvement of ECM, for example. As per the response to point 2, it applies also to endothelial cells." 

Response 1:  Line 476 the two sentences about AAT on tumor microenvironment (such as ECM and angiogenic process) are summary sentences. The specific summaries and examples correspond to the following section 4.4 AAT may effect on inflammatory mediators can influence the angiogenic process;  and section 4.5 AAT can affect tumor cells by regulating the extracellular matrix (ECM) within the TME. 

4.5. AAT can affect tumor cells by regulating the extracellular matrix (ECM) within the TME

The ECM is a complex network comprised of proteins and glycosaminoglycans that serves as a critical scaffold for cellular constituents and significantly influences the TME. This matrix is pivotal in directing tumor cell behavior, including proliferation, invasion, and metastasis, while also protecting tumor cells from apoptosis induced by chemotherapeutic drugs, which leads to drug resistance and tumor resurgence post-treatment[62]. There are various biochemical and structural types of the extracellular matrix (ECM). Specific signals are conveyed to cells by each form and its three-dimensional structure, affecting vital processes such as immune cell migration into inflammatory tissues and immune cell differentiation in the early phases of inflammation[63].

SERPINA1 can affect the remodeling of ECM and cooperate with its components to drive the migration and growth of cancer cells and affect further progression[64, 65]. SERPINA1 enhances communication between fibroblasts and endothelial cells[66, 67] , regulates fibroblast activity by inhibiting ECM-degrading proteases, and has anti-inflammatory properties that affect fibroblast activation. Tuder, R.M. and I. Petrache [68] reviewed that AAT encoded by SERPINA1 may prevent the activation of alveolar macrophages by neutralizing certain enzymes in the coagulation process, especially thrombin and plasmin. This stops the activation of protease-activated receptors (PAR) caused by either cigarette smoke or thrombin. AAT could help regulate this process while also protecting the lungs in other ways[26]. By altering the inflammatory milieu within the TME, SERPINA1 exerts influence on fibroblast behavior, thus affecting the physical and biochemical characteristics of the ECM. The potential for these ECM changes to impact tumor cell invasion, metastasis, and immune cell accessibility to tumor cells highlights the complex function of SERPINA1 in cancer pathophysiology.

4.4. AAT may effect on inflammatory mediators can influence the angiogenic process

Endothelial cells line the inside of blood vessels and play a crucial role in angiogenesis, the formation of new blood vessels. In the early stages of cancer, tumor cells depend on passive diffusion for nutrient acquisition and gas exchange. However, as the tumor grows and seeks to spread, it induces angiogenesis to develop a new vascular supply. The protein AAT influences angiogenesis, which affects protease activity and inflammatory responses, key factors in the angiogenic process. SERPINA1 specifically modulates serine protease activity, reducing pro-angiogenic and pro-inflammatory responses when PAR-2 is activated in endothelial cells[59]. In gastrointestinal diseases, increased SERPINA1 expression is linked to better endothelial cell survival, vascular abnormalities, new vessel growth, and higher vascular permeability[60]. SERPINA1 plays a key role in the communication between endothelial and tumor cells within the TME, impacting various aspects of cancer development and progression[61]. Thus, SERPINA1 is integral to the crosstalk between endothelial cells and neoplasms, influencing myriad facets of cancer pathogenesis and evolution.